# MicroRNA-30b Is Both Necessary and Sufficient for Interleukin-21 Receptor-Mediated Angiogenesis in Experimental Peripheral Arterial Disease

**DOI:** 10.3390/ijms23010271

**Published:** 2021-12-27

**Authors:** Tao Wang, Liang Yang, Mingjie Yuan, Charles R. Farber, Rosanne Spolski, Warren J. Leonard, Vijay C. Ganta, Brian H. Annex

**Affiliations:** 1State Key Laboratory of Respiratory Diseases, Guangzhou Institute of Respiratory Health, Guangzhou Medical University, Guangzhou 510120, China; taowang@gzhmu.edu.cn; 2Robert M Berne Cardiovascular Research Center, University of Virginia, Charlottesville, VA 22908, USA; yangliang@nankai.edu.cn (L.Y.); yuanmj8341@163.com (M.Y.); vganta@augusta.edu (V.C.G.); 3Department of Pharmacology, Nankai University, Tianjing 300071, China; 4Department of Cardiology, Renmin Hospital of Wuhan University, Wuhan 430070, China; 5Center for Public Health Genomics, University of Virginia, Charlottesville, VA 22908, USA; crf2s@virginia.edu; 6Laboratory of Molecular Immunology and the Immunology Center, National Heart Lung and Blood Institute, National Institutes of Health, Bethesda, MD 20892, USA; spolskir@nhlbi.nih.gov (R.S.); wjl@helix.nih.gov (W.J.L.); 7Vascular Biology Center and Department of Medicine, Augusta University, Augusta, GA 30912, USA

**Keywords:** vascular disease, gene therapy, innate immunity, pre-clinical models

## Abstract

The interleukin-21 receptor (IL-21R) can be upregulated in endothelial cells (EC) from ischemic muscles in mice following hind-limb ischemia (HLI), an experimental peripheral arterial disease (PAD) model, blocking this ligand–receptor pathway-impaired STAT3 activation, angiogenesis, and perfusion recovery. We sought to identify mRNA and microRNA transcripts that were differentially regulated following HLI, based on the ischemic muscle having intact, or reduced, IL-21/IL21R signaling. In this comparison, 200 mRNAs were differentially expressed but only six microRNA (miR)/miR clusters (and among these only miR-30b) were upregulated in EC isolated from ischemic muscle. Next, myoglobin-overexpressing transgenic (MgTG) C57BL/6 mice examined following HLI and IL-21 overexpression displayed greater angiogenesis, better perfusion recovery, and less tissue necrosis, with increased miR-30b expression. In EC cultured under hypoxia serum starvation, knock-down of miR-30b reduced, while overexpression of miR-30b increased IL-21-mediated EC survival and angiogenesis. In Il21r^−/−^ mice following HLI, miR-30b overexpression vs. control improved perfusion recovery, with a reduction of suppressor of cytokine signaling 3, a miR-30b target and negative regulator of STAT3. Together, miR-30b appears both necessary and sufficient for IL21/IL-21R-mediated angiogenesis and may present a new therapeutic option to treat PAD if the IL21R is not available for activation.

## 1. Introduction

Peripheral arterial disease (PAD) is caused by atherosclerosis, leading to occlusions of the arteries that supply the lower extremities; PAD affects more than 12 million people in the U.S. and millions more worldwide [1,2,3,4]. The primary problem in PAD is reduced blood flow to the leg(s), and in most patients with symptomatic, advanced PAD, there is a total blockage along the sole major arterial pathway that supplies blood to the leg, and the amount of blood flow to the lower extremity becomes entirely dependent on the extent of ischemia-induced angiogenesis to allow ambulation and avoid limb amputation [5,6].

Surgically induced hind-limb ischemia (HLI) is a widely used pre-clinical model of PAD in mice [7,8,9]. We utilized an unbiased discovery strategy that was based on known differences in the extent of perfusion recovery across inbred mouse strains following HLI, and showed that the IL-21R was significantly upregulated on ischemic endothelium in-vivo from a mouse strain that showed good perfusion recovery after HLI but not in a strain that showed poor perfusion recovery [10]. In addition, in IL21R-deficient (*Il21r*^−/−^) mice on a C57Bl6 background vs. C57Bl6 wild-type (WT), and in WT mice treated with IL-21R-Fc chimera (absorbs and eliminates available IL21) vs. control IgG, there was less STAT3 activation, less angiogenesis, and poorer perfusion recovery in the ischemic limb after HLI [10]. Further, studies in cultured endothelial cells (ECs) demonstrated that IL-21 enhanced EC survival and tube formation under the PAD-relevant hypoxia serum starvation (HSS) condition, however, in EC, under normoxia IL-21 addition, it did not cause an angiogenic effect or STAT3 activation [10].

When considered as a potential therapeutic for PAD, two problems regarding the IL21/IL21R pathway became readily apparent. First, we had shown that the angiogenic effects of IL21 were receptor dependent [10]. Although we reported that, as a group, muscle biopsies from the ischemic limb of patients with one form of PAD had higher levels of the IL21R on EC when compared to controls, the data showed overlap in values in individuals in each group and the study was specific for those with intermittent claudication [11]. Therefore, we could not be certain the IL21R would always be upregulated and for activation in all patients with PAD. Second, whether administration of the ligand could be achieved without toxicity was unknown [12]. Studies on IL-21/IL-21R-signaling in vitro and in vivo have identified a number of downstream gene-expression changes based on study conditions [12,13,14,15,16]. With little to no data available on the IL21/IL21R pathway in PAD, we performed RNA sequencing (RNA-Seq)-based analysis of whole transcriptomes from ischemic muscle with or without IL-21/IL-21R-pathway interruption, using a soluble IL-21R-Fc chimera as described [10]. We then examined an informative time point 7 days after HLI, because the extent of perfusion recovery was comparable between the two groups; this is a strategy we used in several other publications [10,17,18,19]. Though 200 mRNA were differentially expressed, only a few microRNAs (miRs) clusters were changed by IL-21/IL-21R-pathway interruption. We then proceeded to validate the data from the RNA-seq and examined the expression of the identified miRs in the endothelial cells isolated from ischemic muscle and found that miR-30b was the only miR of the group modulated by IL-21. We went on to show that miR-30b is both necessary and sufficient for IL-21/IL-21R hypoxic-induced angiogenesis and identified a potential gene target for miR-30b.

## 2. Results

### 2.1. Identification of IL-21R Pathway Transcripts following Experimental PAD

Of the 32,062 transcripts quantified by RNA-seq from whole muscle, 200 mRNA were significantly different (q < 0.1) in animals treated with IL-21R-Fc vs. control IgG at 7 days after HLI (Appendix A). Of these transcripts, 131 mRNAs were significantly downregulated, and 69 mRNAs were significantly upregulated. Though a large number of mRNAs were differentially regulated, only six microRNA (miR)/miR clusters were differentially regulated (Table 1). Of these six miR clusters, miR-343 and miR-5103 were not detectable by qPCR in the ischemic muscle from either IL-21R-Fc or control group. From the four remaining miR clusters, when transcript levels from IL-21R-Fc group were compared to muscle from mice treated with control IgG, the miR-let7a-2 and miR-100 cluster, miR-30b within the miR-30b/miR-30d cluster, and miR-503 within the miR-503/miR-322/miR-351 showed significantly lower expression levels (0.38 ± 0.18, 0.27 ± 0.03, 0.27 ± 0.05 and 0.53 ± 0.08 fold relative to control group, n = 4/group, *p* < 0.05, Figure 1); miR-3572 showed significantly higher expression levels (1.43 ± 0.06 fold relative to control group, n = 4/group, *p* = 0.003) (Figure 1A); miR-322, miR-351 and miR-30d levels showed no significant difference (Figure 1A).

Since we previously showed that IL-21R was upregulated in the endothelial cells under ischemia in both mouse and human PAD muscle [10,11], we isolated CD31-expressing cells from ischemic hind-limb muscles of *Il21r*^−/−^ or wild-type (*WT)* mice, as described, [10] and measured the expression level of miR-let7a-2, miR-300, miR-30b, miR-503 and miR-3572. MiR-30b was the only of these miRs which showed significant difference in those from *Il21r*^−/−^ mice (0.08 ± 0.04-fold relative to EC from *WT* mice, *p* = 0.04, Figure 1B). miR-3572 was not detectable in endothelial cells. Next, we compared the expression of miR-let7a-2, miR-300, miR-30b, miR-503 and miR3572 in HUVECs cultured under HSS conditions, with or without IL-21 treatment for 24h, and miR-30b was the only one that showed a differential expression level (1.91 ± 0.12-fold to basal, *p* = 0.01, Figure 1B). MiR-3572 was not detectable by qPCR. Interestingly, even though miR-30b is co-localized with miR-30d in the same microRNA cluster, miR-30d was not differentially regulated by qPCR from any of the following: (a) ischemic muscle (1.01 ± 0.12-fold to control, *p* = 0.14, Figure 1B), endothelial cells isolated from ischemic muscle (1.03 ± 0.11-fold to control, *p* = 0.46, Figure 1B) or cultured endothelial cells (1.00 ± 0.16-fold to control, *p* = 0.90, Figure 1C).

### 2.2. IL-21 Overexpression Improves Perfusion Recovery, Increases STAT3 Phosphorylation, and Upregulates miR-30b in the Ischemic Muscle after Hind Limb Ischemia (HLI) in Myoglobin Transgenic Mice

Myoglobin transgenic (MgTG) mice were used to test the effects of IL-21 overexpression following HLI for several reasons. First, compared to wild type (WT) C57BL/6 mice, these mice have impaired angiogenesis and poorer perfusion recovery following HLI [30,31]. Second, this strain was generated on a C57BL/6 background, where we previously showed findings that the IL-21R is upregulated in the C57BL/6 strain [10]. Third, the transgene includes the entire myoglobin gene as well as its striated muscle-specific promoter, so that overexpression is limited to striated muscle myocyte, and the endothelium from these mice is the same as wild-type [30,31].

As predicted, the MgTG mice on the C57BL/6 background showed a 10.6 ± 1.8-fold increase of *Il21r* expression in the ischemic muscle when compared to non-ischemic muscle tissue, 7 days after HLI (Figure 2A). To augment IL-21 expression and IL21R pathway activation, we injected an expression plasmid containing IL-21, or its control plasmid, into the hind-limb muscles. Compared with those receiving control (scrambled sequence) plasmid, MgTG mice receiving IL-21 plasmid showed improved perfusion recovery at day-21 after HLI (67.9% ± 3.3% vs. 57.6% ± 3.2%, *p* = 0.02, Figure 2B) and less tissue necrosis (0 out of 6 vs. 4 out of 7, *p* < 0.01). Confirming the specificity of the findings, *Il21r*^−/−^ mice receiving IL-21 cDNA did not show any changes in tissue necrosis (4 out of 12 vs. 3 out of 10, *p* = 0.86) or perfusion recovery at any of the same time points (54.5% ± 6.4% vs. 58.5% ± 7.1%, *p* = 0.68, Figure 2C). Next, we determined the expression of IL-21 at 10 days after plasmid injection and IL-21 protein expression was significantly higher in the ischemic hind-limb muscle of the mice that received IL-21 cDNA compared with mice that received scramble plasmid in both MgTG mice (IL-21/ERK2, 0.27 ± 0.08 vs. 0.06 ± 0.01, n = 5/group, *p* = 0.02) and *Il21r*^−/−^ mice (IL-21/ERK2, 0.62 ± 0.08 vs. 0.22 ± 0.02, n = 6–7/group, *p* = 0.02) (Figure 2D). In Figure 2E, we showed that the ischemic hind-limb muscle tissue from MgTG mice receiving IL-21 plasmid had a higher capillary density compared with those receiving control plasmid (1.9 ± 0.18 vs. 1.2 ± 0.17, capillaries/fiber, n = 6–7/group, *p* = 0.03) 21 days after HLI.

The IL-21R can signal via STAT1, STAT3, Akt, and ERK1/2 in different conditions [12,15] and we tested for potential differences in activation along with these pathways. Mice that received IL-21-expressing plasmid also had greater STAT3 phosphorylation (p-STAT3/STAT3, 0.72 ± 0.15 vs. 0.35 ± 0.10, n = 5/group, *p* = 0.02) but no change in STAT1 (p-STAT1/STAT1, 0.42 ± 0.04 vs. 0.43 ± 0.05, n = 5/group, *p* = 0.74), Akt (p-Akt/Akt, 0.24 ± 0.01 vs. 0.28 ± 0.01, n = 5/group, *p* = 0.17) or ERK1/2(p-ERK/ERK, 0.42 ± 0.04 vs. 0.36 ± 0.05, n = 5/group, *p* = 0.39) phosphorylation in ischemic muscle 1 day after HLI when compared with mice receiving control plasmid (Figure 3A). Next, we measured miR-30b expression in the ischemic muscle specimen by qPCR and showed a significant higher expression from MgTG mice receiving IL-21 cDNA (2.73 ± 0.41-fold to control, n = 5, *p* = 0.03) (Figure 3B) when compared with those receiving control plasmid 7 days after HLI.

### 2.3. Overexpression of miR-30b Improves Perfusion Recovery in Il21r^−/−^ Mice

To test whether miR-30b modulates the response to HLI, we overexpressed miR-30b in *Il21r*^−/−^ mice using local intramuscular injections of pre-miR-30b overexpression plasmid 3 days before HLI with empty vector (EV) was used as control. Expression of miR-30b was assessed 10 days post plasmid injection and was increased >100 fold in ischemic hind limbs of mice vs. empty vector (Figure 4A). Next, we determined the effect of this gene transfer on perfusion recovery following experimental PAD, as shown in Figure 4B, miR-30b overexpression significantly improved perfusion recovery at day 14 (63.3% ± 3.0% vs. 49.5% ± 3.5%, n = 11~12/group, *p* < 0.01) and day 21 (74.4% ± 4.1% vs. 58.4% ± 4.6%, n = 11~12/group, *p* = 0.016) post HLI. There was a numeric but not statistically significant reduction in hind-limb necrosis rate (2/12 vs. 4/11, *p* = 0.14).

### 2.4. MiR-30b Is Required for IL-21-Mediated Angiogenesis In Vitro

We next altered miR-30b expression in HUVECs and assessed the effects of modulating miR-30b on cellular survival and tube formation. HUVECs transfected with miR-30b mimic had 909 ± 286-fold overexpression of miR-30b (Figure 5A, left), increased cell viability (OD450, 0.35 ± 0.01 vs. 0.27 ± 0.001, n = 8/group, *p* < 0.001, Figure 5A, right) tube formation (tube length, 3351 ± 235 vs. 2395 ± 207 µm/mm^2^, n = 8/group, *p* = 0.01, Figure 5B) under HSS conditions. To investigate whether miR-30b is required for the survival effects of IL-21 treatment in hypoxic endothelial cells, HUVECs were transfected with miR-30b (hsa-miR-30b-5p) inhibitor, resulting in effective knockdown of miR-30b (Figure 5C, left). Compared with control-miR inhibitor, mir-30b inhibitor abrogated the ability of IL-21 to enhance cell viability (OD450: 0.21 ± 0.07 vs. 0.21 ± 0.005, n = 8/group, *p* > 0.05) (Figure 5C, right) and tube formation (Tube length: 2505 ± 111 vs. 2551 ± 101 µm/mm^2^, n = 8/group, *p* > 0.05) (Figure 5D).

### 2.5. MiR-30b Increased STAT3 Phosphorylation and Reduced SOCS3 Expression

Suppressor of cytokine signaling 3 (SOCS3) was an important molecule that regulates STAT3 phosphorylation [32,33], and using Targetscan SOCS3 was a potential target of miR-30b in both human and mice. In mice following HLI, as shown in Figure 6A, we found that SOCS3 protein was lower in miR-30b overexpression vs. empty vector-treated mice at day 3 post HLI (SOCS3/ERK2, 0.49 ± 0.09 vs. 0.04 ± 0.02, n = 4, *p* < 0.01). Consistent with this finding, in HUVECs cultured under HSS conditions, miR-30b augmentation reduced SOCS3 protein expression (SOCS3/ERK2, 0.70 ± 0.05 vs. 0.45 ± 0.03, n = 4/group, *p* = 0.01, Figure 6B). As SOCS3 is a suppressor of STAT3, we also measured the level of STAT3 phosphorylation that is increased with miR-30b overexpression (p-STAT3/STAT3, 0.86 ± 0.08 vs. 0.28 ± 0.07, n = 4/group, *p* = 0.001, Figure 6B). These may suggest that in ischemic endothelial cells, miR-30b increases STAT3 activation by targeting SOCS3.

## 3. Discussion

Having previously shown an unexpected role for the IL21/IL-21R pathway in mediating hypoxia-dependent angiogenesis [10], we now add several new findings regarding this unexpected pathway. First, we show that miR-30b is the only miR differentially regulated based on the presence or absence of IL-21/IL-21R-pathway activation in the hypoxia-dependent angiogenesis that occurs in experimental PAD. Second, the hypoxia dependent IL-21/IL-21R angiogenesis pathway utilizes and requires miR-30b for its context-dependent angiogenic effects. Third, the IL-21/IL-21R pathway following HLI and EC under HSS, involves activation of the STAT3 pathway, and miR-30b reduces SOCS3, which would be predicted to increase STAT3 phosphorylation. Finally, in the presence of IL-21R, IL-21-ligand overexpression improves perfusion recovery, induces therapeutic angiogenesis, and reduces tissue loss in experimental PAD. Together these finding suggest that the IL-21/IL-21R pathway utilizes and appears dependent on the regulation on a single micro-RNA (Figure 7).

To our knowledge, this is the first report connecting miR-30b to the IL-21/IL-21R pathway. Under experimental conditions relevant to PAD, we found that miR-30b was the only miR differentially regulated by the IL-21/IL-21R pathway in hypoxic angiogenesis. Our in vitro studies indicated that miR-30b was required for IL-21-induced angiogenesis in cultured endothelial cells and our in vivo studies indicate that miR-30b overexpression is sufficient to rescue impaired perfusion recovery after HLI in *Il21r*^−/−^ mice. Some information is available on other miRs that are modulated by IL-21/IL-21R. Adoro et al. reported that miR-29 is required for IL-21-induced anti-viral effects in CD4+ T cells [33]; Rasmussen et al. found that miR-155a is required for IL-21-mediated STAT3 phosphorylation in CD4+ T cells from systemic lupus erythematosus (SLE) patients [34]; using microarray analysis, De Cecco et al. found IL-21 regulates CCL17, CD40, DDR1, and PIK3CD expression through miR-663b in chronic lymphocytic leukemia [35]. As we did not find differential expression of miR-29, 155a, or 663b, our data suggests an EC-specific effect of miR-30b on IL-21-induced angiogenesis. While one previous study indicated that miR-30b overexpression in endothelial cells increases vessel number and length in an in vitro sprouting angiogenesis model [36], another study demonstrated that depletion of miR-30b increased capillary-like cord formation in HUVECs under normoxia conditions [37]. These may suggest that the effects of miR-30b on angiogenesis are condition-specific.

It is interesting that our data suggest that a single miR may regulate hypoxia-dependent angiogenesis both in vitro and in vivo environments, in an unexpected biologic pathway that we previously showed involved the phosphorylation/activation of STAT3. A limited number of such examples could be found. For example, secreted from tumors, miR-9 has been shown to increase endothelial cell migration and angiogenesis via targeting SOCS5 and increasing STAT3 phosphorylation [38]. In addition, miR-337, miR-17, miR-20, miR-24 and miR-629 have been reported to regulate STAT3 activation [39,40].

A miR cluster is a set of two or more miRNAs, which are usually co-transcribed, yet we found that only miR-30b is upregulated by IL-21 in hypoxic endothelial cells, but miR-30d was not regulated. In mice, miR-30b is located within a non-coding region on chromosome 15 and is co-localized with miR-30d in the Mirc26 cluster [41], while in humans, miR-30b and miR-30d co-localized within non-coding RNA LOC102723694 on chromosome 8. Recent studies by others also showed that miRs from the same cluster could be regulated at different levels and even in divergent patterns. For example, Knudsen et al. reported that among four miRs in miR-17–92 cluster, miR-17 is the most upregulated in early colon cancer [42]; and another study using global analysis of miR clusters expression between breast tumor and adjacent tissue found that among miR-221/222 clusters, miR-221 is higher but miR-222 is lower in the tumor tissue [43]. The precise mechanisms by which this level of miR expression is controlled will require additional study.

Our data presented in this report is the first to show the therapeutic potential of modulating this IL21/IL21R hypoxia-dependent angiogenesis pathway in PAD. The specificity of the therapeutic effects of IL-21 overexpression on hypoxic angiogenesis and its receptor dependence was demonstrated by the absence of an effect of IL-21 on *Il21*^−/−^ mice. We showed that blocking IL-21 signaling in hypoxic EC resulted in reduced miR-30b expression and IL-21R activation by ligand in ischemic muscle, resulting in increased miR-30b expression. Finally, miR-30b overexpression improves perfusion recovery in *Il21r*^−/−^ mice. MiRs have great potential as therapeutic agents due to their ease of synthesis, stability, and ability of single miR to regulate several genes within a pathway. Modulation of a single miR has been reported to be therapeutic to PAD [44,45,46,47]. Taken together with our findings, these suggest that miR-30b is a potential therapeutic target for PAD subjects and could be an alternative to ligand administration.

Enhanced STAT3 activation is reported in ischemic tissue from a spectrum of ischemic diseases, including stroke and myocardial infarction, and functions as a protective factor to improve the recovery of these diseases [48,49]. In our previous study, IL-21R loss of function in a pre-clinical PAD model resulted in reduced angiogenesis and perfusion recovery through decreased STAT3 activation [10]. Conversely, this study demonstrated that after HLI, IL-21 overexpression increased perfusion recovery as well as STAT3 activation. Furthermore, we found that miR-30b, a molecule which is required for IL-21-induced angiogenesis under ischemia, also increases STAT3 activation with reduced SOCS3 expression. These data may suggest that miR-30b is in part or fully involved in IL-21-mediated STAT3 phosphorylation by targeting SOCS3 in hypoxia-dependent angiogenesis. In summary, we have demonstrated that a novel angiogenic IL-21/IL-21R/miR-30b/STAT3 promotes endothelial cell survival under ischemia in PAD.

## 4. Material and Methods

### 4.1. Hind-Limb Ischemia (HLI) Model, Perfusion Recovery, and In Vitro Transfection

Methods for the HLI model were performed as previously described [10,20,21]. Briefly, unilateral femoral-artery ligation and excision were performed on the left side of mice. Perfusion flow in the ischemic and contralateral non-ischemic limbs was measured on day 0, 7, 14, and 21 post-HLI with laser Doppler perfusion imaging system (Perimed, Inc., Ardmore, PA, USA), as described previously [20,21]. Perfusion was expressed as the ratio of the left (ischemic) to right (non-ischemic) hind limb. Limb necrosis was determined with ordinal values, as described [22].

For IL-21 overexpression, a full-length murine IL-21 cDNA was purchased from Open Biosystems (Huntsville, AL, USA). The cDNA was directionally cloned in the p-cytomegalovirus (CMV)-TnT (Promega Corp., Madison, WI, USA) plasmid that directs high expression from the CMV promoter upon transfection into mammalian cells. Resulting clones were sequenced to verify that there were no errors introduced during the PCR cloning reactions. Mice were grouped to receive pCMV-TnT plasmid vectors delivering IL-21 cDNA or a scrambled DNA sequence as a control. For mmu-miR-30b transfection, expression plasmid of mmu-miR-30b-5p (pCMV-miR-30b) and empty vector (EV) control were purchased from Origene (Rockville, MD, USA). Mice were grouped to receive pCMV-miR-30b plasmid vector or EV. Plasmids were transfected into mouse hind-limbs via electric, pulse-mediated gene transfer as described previously [21]. Mice were allowed to recover for 3 days before use in experimental PAD studies. Animal studies were approved by the Institutional Animal Care and Use Committee and conformed to the Guide for the Care and Use of Laboratory Animals published by the US National Institutes of Health.

### 4.2. Identification of Transcripts Regulated by IL-21R following HLI

Male C57BL/6J mice (14 to 18 weeks of age) were purchased from the Jackson Laboratory (Bar Harbor, ME) for IL-21R-Fc treatment. A mouse IL-21R Fc chimera (IL-21R-Fc) fusion protein was prepared in the Protein Expression Laboratory, National Cancer Institute and was used to neutralize IL-21 [23]. IL-21R-Fc were injected at a dose of 0.2 mg/mouse intra-peritoneally to C57BL/6J mice immediately (0) and 1, 3 and 5 days after surgical HLI (n = 4), as previously described [10], and an equivalent dose of mouse IgG (Sigma-Aldrich, St. Louis, MO, USA) was used in the control group (n = 4). Seven days after experimental PAD, mice were euthanized, and gastrocnemius muscles from the mice with IL-21R-Fc or control antibody treatment were harvested for RNA isolation and sequencing.

### 4.3. RNA Isolation and Sequencing

Total RNA was isolated from the ischemic gastrocnemius muscle (n = 4/group) using PureLink^®^ RNA Kit (Life Technology, Grand Island, NY, USA), as previously described [10,19]. The quality of RNA was monitored by RNA gel electrophoresis following the manufacturer’s instructions. RNA sequencing (RNA-Seq) was performed as described previously [24]. Briefly, RNA-seq libraries were constructed from 1 µg of total RNA using the TruSeq Stranded Total RNA prep kit (Illumina, San Diego, CA, USA). The resulting barcoded libraries were sequenced on an Illumina NextSeq 500 using 150-cycle High Output Kits (Illumina). Quality control of the resulting sequence data was performed using FastQC (http://www.bioinformatics.babraham.ac.uk/projects/fastqc (accessed on 28 November 2021)). All data were analyzed using Illumina BaseSpace applications. Sequencing reads were aligned to the mouse mm10 reference transcriptome and using the TopHat Alignment v1.0.0 application [25]. Expression values (FPKM; Fragments Per Kilobase of transcript per Million mapped reads) were then generated for each transcript and differentially expressed transcripts (at a false discovery rate (FDR, q-value) of <0.1) were identified using Cufflinks Assembly and Differential Expression v1.1.0 application.

### 4.4. Quantitative RT-PCR (qPCR)

To validate data from the RNA-seq, qPCR was performed using primer/probes from Applied Biosystems (Foster City, CA, USA), similar to methods described previously [10,26]. Small nucleolar RNA MBII-202 (sno202) transcript was used for normalization of miRs loading, and hypoxanthine-guanine phosphoribosyltransferase-1 (*Hprt-1*) was used for normalization of mRNA loading. The 2^−ΔΔCt^ method was used to calculate fold changes as described previously [27,28].

### 4.5. Protein Analysis

Levels of selected target proteins were analyzed by Western blotting using antibodies to total and phosphorylated (p)-signal transducer and activator of transcription (STAT) 1 (p-Y701), p-STAT3 (p-Y705), STAT3 (Cell Signaling, Danvers, MA, USA) and Suppressor of cytokine signaling 3 (SOCS3, Cat# 626601, Biolegend, San Diego, CA, USA), as previously described [29]. Western blots were analyzed by Odyssey Infrared Imaging System (LI-COR Biosciences, Lincoln, NE, USA) and quantified by ImageJ software (National Institute of Health, Bethesda, MD, USA).

### 4.6. In Vitro Transfection, Cellular Viability and Angiogenesis Assay

Pooled human umbilical vein endothelial cells (HUVEC) were purchased (Cell Applications Inc., San Diego, CA, USA). To mimic the endothelial cells under ischemic condition in HLI models, HUVECs were exposed to HSS conditions, as previously described. In vitro transfection was performed to overexpress/knockdown using miR-30b (hsa-miR-30b-5p) mimic or inhibitor, as previously described [18].

For cellular viability studies, HUVECs were plated in a 96-well plate at a density of 1 × 10^4^ cells/well (n = 8/group), and then cultured under HSS conditions for 48 h. At the end of the incubation, cell viability was assessed using tetrazolium dye incorporation (BioVision, Milpitas, CA, USA). In vitro angiogenesis assay was performed as previously described [10,17,18,19]. Briefly, after exposure to HSS conditions for 24 h, transfected HUVECs were plated at a density of 1 × 10^4^ cells/well on 96-well dishes which were coated with growth factor-reduced Matrigel (BD Biosciences, San Jose, CA, USA), and then exposed to HSS conditions for 6 h with rhIL-21 (50 ng/mL) or with vehicle alone to assess tube formation. Each condition was performed in 6 wells. The degree of tube formation was determined by measuring the length of the tubes and the number of loops from each well under 40× magnification using the online WimTube application module (Wimasis GmbH, Munich, Germany). Each experiment was repeated on at least 2 different batches of HUVECs.

### 4.7. Data Analysis and Statistics

Statistical analysis was performed with GraphPad Prism software. An unpaired *t* test was used for comparison between 2 groups, and comparisons in experiments with ≥3 groups were performed with one-way ANOVA and the Tukey post hoc test. Differences in necrosis rate after HLI were analyzed by chi-square test. Statistical significance was set at *p* < 0.05. Statistical methods of RNA-seq data are described in above in Section 2.3.

## Figures and Tables

**Figure 1 ijms-23-00271-f001:**
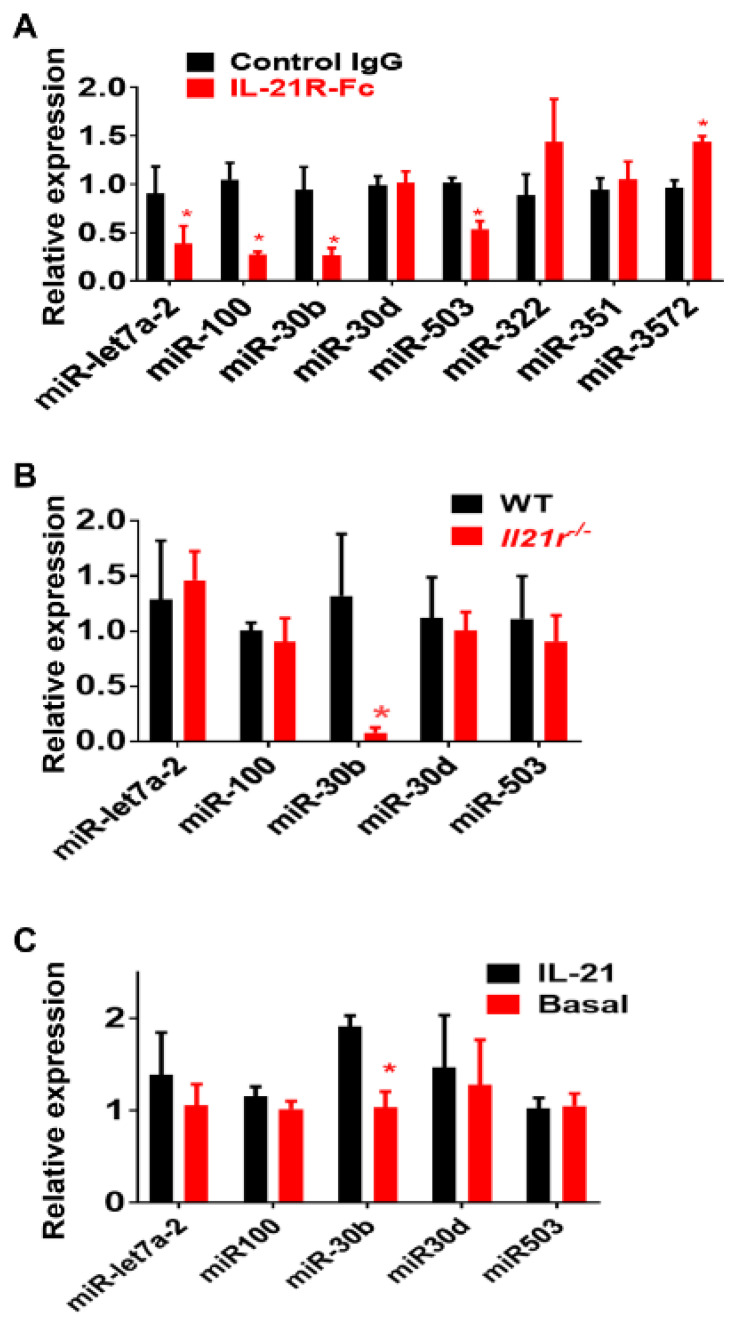
(**A**) Validation of RNA-sequencing analysis by real-time qPCR for microRNAs (miR) regulated by IL-21R-Fc treatment in ischemic muscle 7 days after hind-limb ischemia (HLI). When transcript levels were compared to muscle from mice treated with control IgG, miR100, miR-let7a-2, miR30b in miR30b/miR30d cluster, and miR503 in miR503/miR322/miR351 cluster showed significantly lower expression levels; 3572 showed significantly higher expression levels; miR322, miR351 and miR30d level showed no significant difference. (**B**) Expression of miR-let7a-2, miR-100, miR-30b, miR-30d and miR-503 in CD31 (endothelial cell marker)-enriched fraction of cells isolated from ischemic hindlimbs of wild-type (*WT*) and *Il21r*^−/−^ mice were quantified by qPCR, miR30b showed significant lower expression in endothelial cells from *Il21r*^−/−^ mice when compared to endothelial cells from WT mice (n = 4/group). However, the other four microRNAs did not show any significant difference between the two groups. (**C**) In HUVECs cultured under hypoxia serum starvation (HSS) (used to mimic ischemic muscle in vivo), 24 h treatment of IL-21 increased miR30b expression (n = 4/group, * *p* < 0.05), but did not regulate the expression of the other four microRNAs. * *p* < 0.05. Fold changes of each miR expression were calculated based on the average ΔCt value of the control group (for details, see material and methods). Data represent mean ± SEM; n = 4 per group. Control IgG indicates ischemic gastrocnemius muscle from the mice treated with control IgG, IL-21R-Fc ischemic gastrocnemius muscle from the mice treated with IL-21R-Fc.

**Figure 2 ijms-23-00271-f002:**
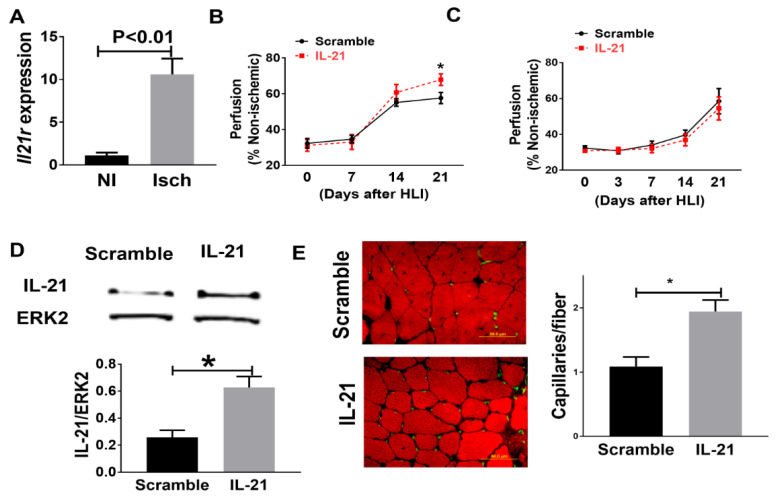
(**A**) Interleukin-21receptor (IL-21R) expression is upregulated in the ischemic limb from MgTG mice when compared to the non-ischemic limb. (**B**) Data from Laser Doppler Perfusion imaging (LDPI) showed that IL-21 overexpression improved perfusion recovery in MgTG mice at day 21 (n = 6~7/group). (**C**) IL-21 overexpression did not change perfusion recovery in IL-21R knockout (KO) mice at any of the selected time point. (**D**) IL-21 plasmid transfection showed higher IL-21 protein level in the ischemic limb muscle from *Il21r*^−/−^ mice muscle. (**E**) At day 21 after HLI, ischemic gastrocnemius muscle from MgTG mice that received IL-21 overexpression plasmid showed significant higher capillary density than mice that received control plasmid (n = 6~7/group). Average capillaries per muscle fiber: 1.9 ± 0.18 vs. 1.1 ± 0.17, *p* = 0.03. Data = mean ± SEM. * *p* < 0.05.

**Figure 3 ijms-23-00271-f003:**
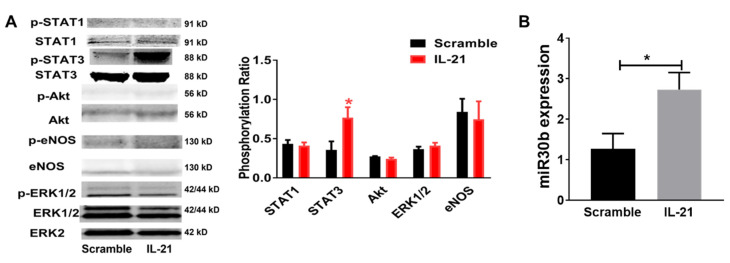
STAT3 phosphorylation and miR-30b expression were upregulated by IL-21 in the ischemic muscle in HLI model. (**A**) Western blots were probed for STAT1, p-STAT1, STAT3, p-STAT3, Akt, p-Akt, ERK1/2 and Erk1/2 with only ERK2 shown as loading control, IL-21 overexpression increases STAT3 phosphorylation in ischemic muscle 1 day after HLI (p-STAT3/STAT3) but shows no change in STAT1, Akt, eNOS or ERK1/2 phosphorylation. (**B**) miR-30b expression is upregulated by IL-21 overexpression in ischemic muscle 7 days after HLI, quantitative normalization of microRNA in each sample was performed using expression of small nucleolar RNA MBII-202 (sno202) as an internal control. Data = mean ± SEM. N = 4–5/group, * *p* < 0.05, IL-21 indicates IL-21 overexpression plasmid; Scramble indicates control plasmid with scramble sequence.

**Figure 4 ijms-23-00271-f004:**
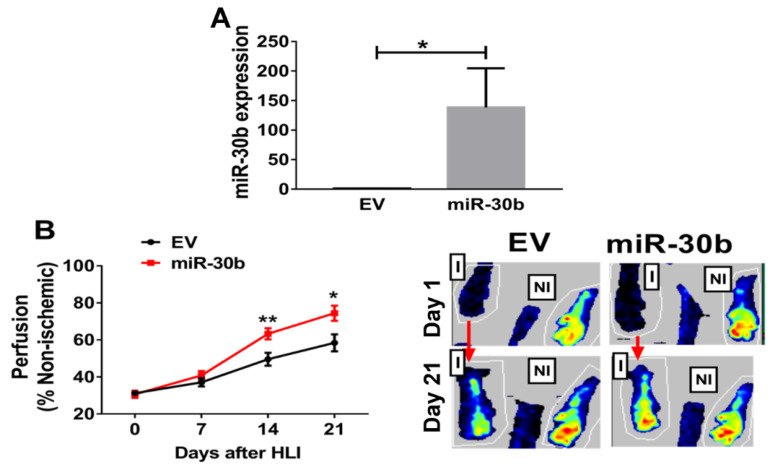
Augmentation of miR-30b expression in *Il21r*^−/−^ mice improved perfusion recovery following HLI. (**A**) Higher miR-30b is expressed in hind limbs that received pCMV-miR-30b compared with those that received control plasmid (empty vector, EV), 10 days post treatment (n = 5/group, *p* = 0.01). (**B**) *Il21r*^−/−^ mice in C57BL/6 background with miR-30b overexpression in the ischemic limb showed significantly better perfusion recovery at days 14 (*p* < 0.01) and 21 (*p* = 0.016) after HLI (n = 11–12 per group). EV indicates empty vector plasmid transfection; miR-30b indicates mice with miR-30b overexpression in the ischemic limb. Data represent mean ± SEM.I = ischemic limb, NI = non-ischemic limb, * *p* < 0.05, ** *p* < 0.01.

**Figure 5 ijms-23-00271-f005:**
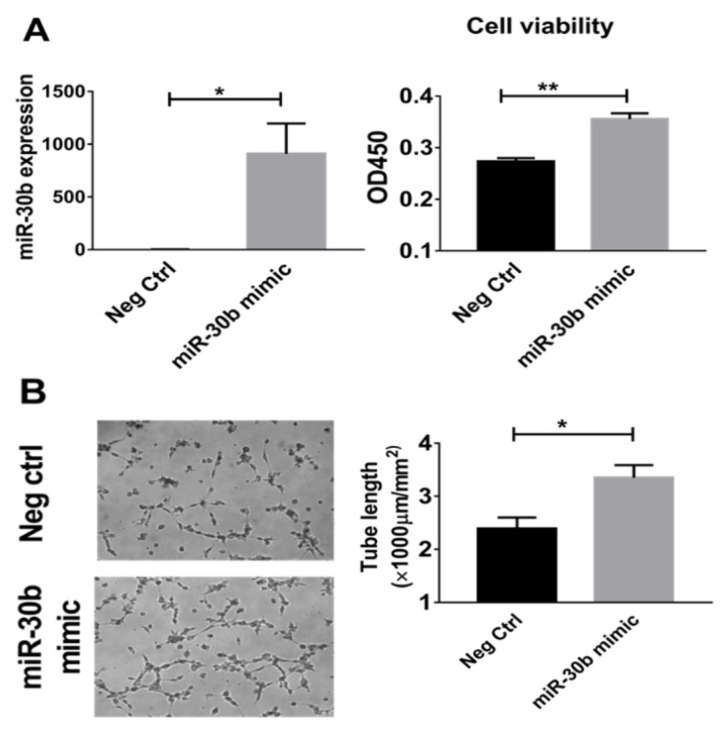
Effects of miR30b modulation in IL-21-mediated endothelial cell survival and angiogenesis under HSS conditions. (**A**) When miR-30b was overexpressed by miR30b mimic transfection in cultured HUVECs (left), more viable cells were detected 48 h after cultured under HSS conditions (right). (**B**) 48 h after transfection with miR-mimic negative control (neg ctrl) or miR30b mimic, HUVECs were plated in Matrigel with reduced growth factor and incubated for 6 h in basal medium. MiR-30b mimic–treated HUVECs showed enhanced tube formation, which was quantified as the tube length per area (bar graph). (**C**,**D**) IL-21 treatment (50 ng/mL) increased HUVEC cell viability and tube formation under HSS conditions, the IL-21 effects on HUVECs viability and tube formation were inhibited when miR30b was knocked down using miR30b inhibitor (C left). Cell viability assay is based on the cleavage of the tetrazolium salt to formazan by cellular mitochondrial dehydrogenase, OD450 indicates optical density at 450 nm. Neg ctrl = negative control RNA for microRNA inhibitor/mimic. Data represent mean ± SEM. All the above data are representative of two to three separate batches of HUVECs, n = 8 to 12 samples per group. * *p* < 0.05, ** *p* < 0.01, NS indicates non-significance.

**Figure 6 ijms-23-00271-f006:**
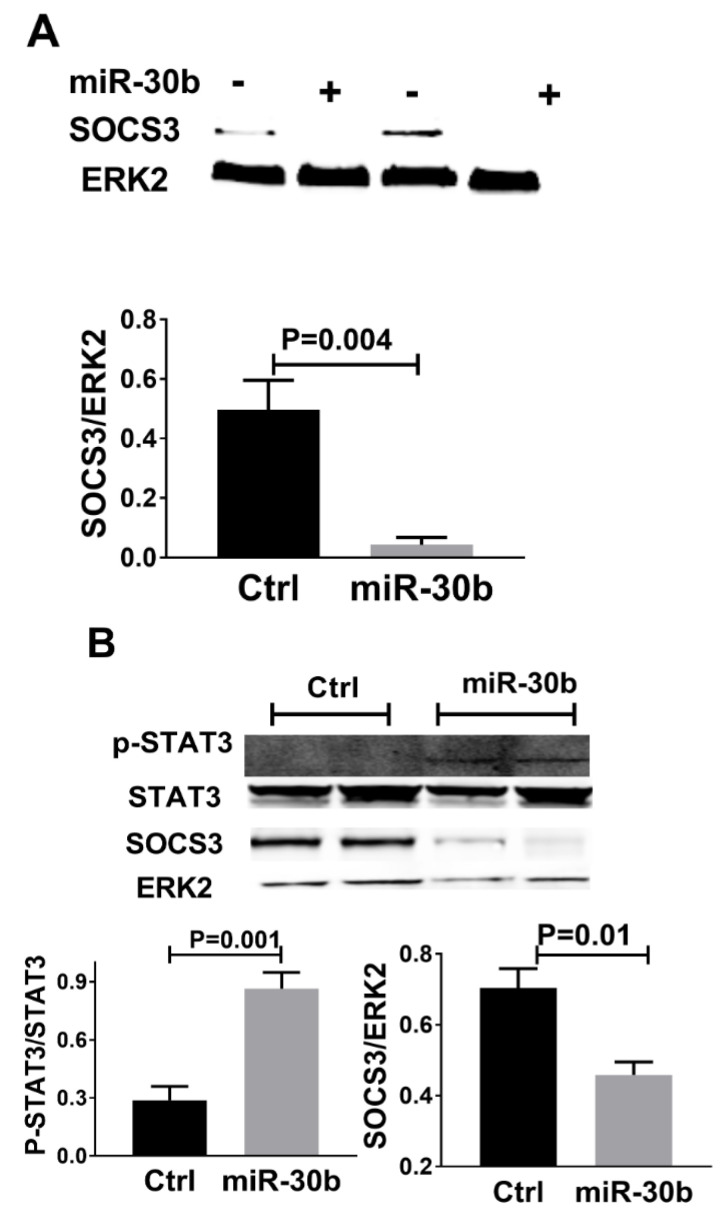
Protein changes in ischemic muscles with modulation of miR-30b in vivo in *Il21r*^−/−^ mice and in vitro human umbilical vein endothelial cells (HUVECs) cultured under hypoxia serum starvation (HSS) conditions. (**A**) Protein isolated from muscle harvested from *Il21r*^−/−^ mice with empty vector or the miR-30b overexpressing plasmid day-3 following HLI treated. Western blots were probed for SOCS3 and Erk1/2 with only ERK2 shown, SOCS3 expression level was significantly downregulated with miR-30b overexpression (*p* < 0.01). (**B**) Overexpression of miR-30b by miR-30b mimic in HUVECs cultured under HSS for 24 h resulted in increased STAT3 phosphorylation (p-STAT3/STAT3, *p* = 0.001) and reduced SOCS3 expression (SOCS3/ERK2, *p* = 0.01). Data represent mean ± SEM., miR-30b indicates miR-30 overexpression by pCMV-miR-30b plasmid in vivo or miR-30b mimic in vitro, Ctrl indicates negative control empty vector for miR-30b overexpression in vivo and negative control RNA for miR-30b mimic in vitro. n = 5/group.

**Figure 7 ijms-23-00271-f007:**
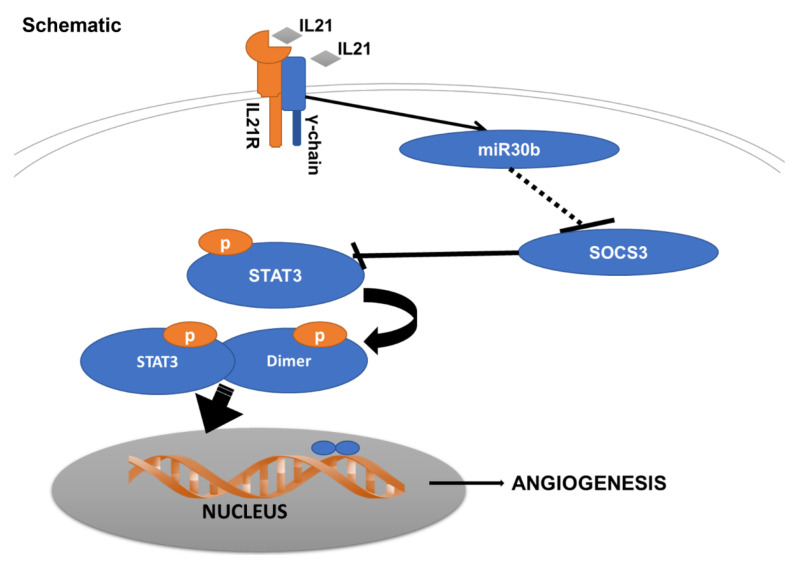
A schematic of proposed IL-21/IL-21R/SOCS3/STAT3 signal in ischemic endothelial cell in PAD conditions such as hypoxia serum starvation or in ischemic leg muscle with ligand (IL21) overexpression. Ligand (small green box) binds to the IL21R (red), which interacts with the common gamma chain (blue) on the cell surface. This complex activation upregulates miR-30b, leading to reduced SOCS3 expression, and results in increased STAT3 phosphorylation and greater amounts of angiogenesis.

**Table 1 ijms-23-00271-t001:** MicroRNA (miR) and miR clusters that are differentially regulated by IL-21R-Fc in the ischemic muscle based on RNA-seq analysis. q-value indicates FDR adjusted p-value of RNA-seq data.

miRs	Fold	q-Value
miR-343	0.03	0.01
miR-5103	0.06	0.01
miR-100, miR-let7a-2	0.17	0.01
miR-30b, miR-30d	0.20	0.01
miR-322,miR-351,miR-503	0.31	0.01
miR-3572	10.08	0.01

## Data Availability

Data will be made available upon reasonable request to the corresponding author.

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
