# Peer review of "MicroRNA-30b Is Both Necessary and Sufficient for Interleukin-21 Receptor-Mediated Angiogenesis in Experimental Peripheral Arterial Disease"

_ijms, 2021, doi:10.3390/ijms23010271_

Round 1

Reviewer 1 Report

The authors have shown intersting results regarding the involvement or conection of miR-30b with the IL-21/IL21R pathway in a HLI model.

Overall, I consider the results interesting and well presented, but I think several things should be checked by the authors:

Perhaps, mmu-miR-30b should be used when talking about the mouse miRNA.

Introduction:

The first sentence must be checked, consider adding commas after atherosclerosis and extremities or a dot after extremities (?)

Results:

Page 6 of 22. Results

The sentence “IL-21 overexpression improves perfusin recovery, increases STAT3 phosphorylation….” In the middle of the text, is it a heading? It´s not clear why is it here if is not a sub-heading. Please, check.

The same after figure 2.

The sentence The IL-21R can signal via STAT1, STAT3, Akt…., is it a sub-heading?

In section 3.2. There are different font sices, please, check, from ---as shown in figure 4B…. 

Figure 6. Could the authors indicate the number of replicates (n) used to perform the statistics?

Author Response

Reviewer-1:
Comments and Suggestions for Authors are in italics and Responses are in bold:
The authors have shown intersting results regarding the involvement or conection of miR-30b with the IL-21/IL21R pathway in a HLI model.
Overall, I consider the results interesting and well presented, but I think several things should be checked by the authors:”
Response: We appreciate the favorable comments. No response is required.
Perhaps, mmu-miR-30b should be used when talking about the mouse miRNA. Response: We reconfirmed that the mouse and human miR-30b sequence is identical. Studies in the manuscript included both murine and human material. Nevertheless, we made the changes as suggested.
Introduction:
The first sentence must be checked, consider adding commas after atherosclerosis and extremities or a dot
after extremities (?)
Response: We apologize for missing this in our final version. This error has been corrected.
Results:
Page 6 of 22. Results
The sentence “IL-21 overexpression improves perfusin recovery, increases STAT3 phosphorylation….” In the middle of the text, is it a heading? It´s not clear why is it here if is not a sub-heading. Please, check.
The same after figure 2.
The sentence The IL-21R can signal via STAT1, STAT3, Akt…., is it a sub-heading?
In section 3.2. There are different font sices, please, check, from ---as shown in figure 4B….
Figure 6. Could the authors indicate the number of replicates (n) used to perform the statistics?
Response: We apologize for the lack of clarity in the above areas. The corrections have been made and the number of replicates added.

Reviewer 2 Report

Wang et al studied the function of microRNA-30b is both necessary and sufficient for interleukin-21 receptor-mediated angiogenesis in experimental peripheral arterial disease. 
This study is a well-thought-out, designed study.
Results: 
1. Figure 3. Western blots of p-STAT1 and p-eNOS were very poor. The only background was present rather than the band. 
2. Figure 3. marking of (A) and (B) missing
3. Figure 3. Details about p-STAT1 and p-eNOS missing in legends. 
The experimental well discussed. 

Author Response

Comments and Suggestions for Authors are in italics and Responses are in bold:
Wang et al studied the function of microRNA-30b is both necessary and sufficient for interleukin-21 receptor-mediated angiogenesis in experimental peripheral arterial disease. This study is a well-thought-out, designed study.
Response: We appreciate the favorable comments. No response specific response is required
“Results:
1. Figure 3. Western blots of p-STAT1 and p-eNOS were very poor. The only background was present rather than the band.
2. Figure 3. marking of (A) and (B) missing
3. Figure 3. Details about p-STAT1 and p-eNOS missing in legends.
The experimental well discussed.”
Response: We apologize for the lack of clarity in the above areas. These corrections have been made in the track changed versions.
